# Optimistic Optimization of a Brownian

**Jean-Bastien Grill**     **Michal Valko**     **Rémi Munos**
SequeL team, INRIA Lille - Nord Europe, France   *and*   DeepMind Paris, France
jbgrill@google.com     michal.valko@inria.fr     munos@google.com

## Abstract

We address the problem of optimizing a Brownian motion. We consider a (random) realization $W$ of a Brownian motion with input space in $[0, 1]$. Given $W$, our goal is to return an $\varepsilon$-approximation of its maximum using the smallest possible number of function evaluations, the *sample complexity* of the algorithm. We provide an algorithm with sample complexity of order $\log^2(1/\varepsilon)$. This improves over previous results of Al-Mharmah and Calvin (1996) and Calvin et al. (2017) which provided only polynomial rates. Our algorithm is adaptive—each query depends on previous values—and is an instance of the *optimism-in-the-face-of-uncertainty* principle.

## 1  Introduction to sample-efficient Brownian optimization

We are interested in optimizing a sample of a standard Brownian motion on $[0, 1]$, denoted by $W$. More precisely, we want to sequentially select query points $t_n \in [0, 1]$, observe $W(t_n)$, and decide when to stop to return a point $\widehat{t}$ and its value $\widehat{M} = W(\widehat{t})$ in order to well approximate its maximum $M \triangleq \sup_{t \in [0,1]} W(t)$. The evaluations $t_n$ can be chosen adaptively as a function of previously observed values $W(t_1), ..., W(t_{n-1})$. Given an $\varepsilon > 0$, our goal is to stop evaluating the function as early as possible while still being able to return $\widehat{t}$ such that with probability at least $1 - \varepsilon$, $M - W(\widehat{t}) \leq \varepsilon$. The number of function evaluations required by the algorithm to achieve this $\varepsilon$-approximation of the maximum defines the *sample-complexity*.

**Motivation**  There are two types of situations where this problem is useful. The first type is when the random sample function $W$ (drawn from the random process) *already exists prior to the optimization*. Either it has been generated before the optimization starts and the queries correspond to reading values of the function already stored somewhere. For example, financial stocks are stored at a high temporal resolution and we want to retrieve the maximum of a stock using a small number of memory queries. Alternatively, the process physically exists and the queries correspond to *measuring it*.

Another situation is when *the function does not exist prior to the optimization* but is built simultaneously as it is optimized. In other words, observing the function actually *creates it*. An application of this is when we want to return a sample of the maximum (and the location of the maximum) of a Brownian motion conditioned on a set of already observed values. For example, in Bayesian optimization for Gaussian processes, a technique called *Thomson sampling* (Thompson, 1933; Chapelle and Li, 2011; Russo et al., 2018; Basu and Ghosh, 2018) requires returning the maximum of a sampled function drawn from the posterior distribution. The problem considered in the present paper can be seen as a way to *approximately perform this step in a computationally efficient way* when this Gaussian process is a Brownian motion.

Moreover, even though our algorithm comes from the ideas of learning theory, it has applications beyond it. For instance, in order to computationally sample a solution of a stochastic differential equation, Hefter and Herzwurm (2017) express its solution as a function of the Brownian motion $W$ and its running minimum. They then need, as a subroutine, an algorithm for the optimization of Brownian motion to compute its running minimum. We are giving them that and it is light-speed fast.

**Prior work** Al-Mharmah and Calvin (1996) provide a *non-adaptive* method to optimize a Brownian motion. They prove that their method is optimal among all non-adaptive methods and their sample complexity is polynomial of order $1/\sqrt{\varepsilon}$. More recently, Calvin et al. (2017) provided an *adaptive* algorithm with a sample complexity lower than any polynomial rate showing that adaptability to previous samples yields a significant algorithmic improvement. Yet their result does not guarantee a better rate than a *polynomial* one.

**Our contribution** We introduce the algorithm `OOB` = optimistic optimization of the Brownian motion. It uses the *optimism-in-face-of-uncertainty* apparatus: Given $n-1$ points already evaluated, we define a set of functions $\mathcal{U}_n$ in which $W$ lies with high probability. We then select the next query point $t_n$ where the maximum of the most optimistic function of $\mathcal{U}_n$ is reached: $t_n \triangleq \arg\max_{t \in [0,1]} \max_{f \in \mathcal{U}_n} f(t)$. This begets a simple algorithm that requires an expected number of queries of the order of $\log^2(1/\varepsilon)$ to return an $\varepsilon$-approximation of the maximum, with probability at least $1 - \varepsilon$ w.r.t. the random sample of the Brownian motion. Therefore, our sample complexity is *better than any polynomial rate*.

**Solving an open problem** Munos (2011) provided sample complexity results for optimizing any function $f$ characterized by two coefficients $(d, C)$ where $d$ is the near-optimality dimension and $C$ the corresponding constant (see his Definition 3.1). It is defined as the smallest $d \geq 0$ such that there exists a semi-metric $\ell$ and a constant $C > 0$, such that, for all $\varepsilon > 0$, the maximal number of disjoint $\ell$-balls of radius $\mathcal{O}(\varepsilon)$ with center in $\{x, f(x) \geq \sup_x f(x) - \varepsilon\}$ is less than $C\varepsilon^{-d}$. Under the assumption that $f$ is locally (around one global maximum) one-sided Lipschitz with respect to $\ell$ (see his Assumption 2), he proved that for a function $f$ characterized by $(d = 0, C)$, his `DOO` algorithm has a sample complexity of $\mathcal{O}(C\log(1/\varepsilon))$, whereas for a function characterized by $(d > 0, C)$, the sample complexity of `DOO` is $\mathcal{O}(C/\varepsilon^d)$. Our result answers a question he raised: *What is the near-optimality dimension of a Brownian-motion?* The Brownian motion being a stochastic process, this quantity is a random variable so we consider the number of disjoint balls in expectation. We show that for any $\varepsilon$, there exists some particular metric $\ell_\varepsilon$ such that the Brownian motion $W$ is $\ell_\varepsilon$-Lipschitz with probability $1 - \varepsilon$, and there exists a constant $C(\varepsilon) = \mathcal{O}(\log(1/\varepsilon))$ such that $(d = 0, C(\varepsilon))$ characterizes the Brownian motion. However, there exists no constant $C < \infty$ independent of $\varepsilon$ such that $(d = 0, C)$ characterizes the Brownian motion. Therefore, we solved this open problem. Our answer is compatible with our result that our algorithm has a sample complexity of $\mathcal{O}(\log^2(1/\varepsilon))$.

## 2 New algorithm for Brownian optimization

Our algorithm `OOB` is a version of `DOO` (Munos, 2011) with a *modified upper bound* on the function, in order to be able to optimize stochastic processes. Consider the points $t_1 < t_1 < ... < t_n$ evaluated so far and $t_0 = 0$. `OOB` defines an upper confidence bound $B_{[t_i, t_{i+1}]}$ for each interval $[t_i, t_{i+1}]$ with $i \in \{0, ..., n-1\}$ and samples $W$ in the middle of the interval with the highest upper-confidence bound. Algorithm 1 reveals its pseudo-code.

---

**Algorithm 1 `OOB` algorithm**

1: **Input**: $\varepsilon$
2: **Init**: $\mathcal{I} \leftarrow \{[0, 1]\}, t_1 = W(1)$
3: **for** $i = 2, 3, 4, \ldots$ **do**
4:    $[a, b] \in \arg\max_{I \in \mathcal{I}} B_I$ {*break ties arbitrarily*}
5:    **if** $\eta_\varepsilon(b - a) \leq \varepsilon$ **then**
6:      **break**
7:    **end if**
8:    $t_i \leftarrow W\left(\frac{a+b}{2}\right)$
9:    $\mathcal{I} \leftarrow \{\mathcal{I} \cup [a, \frac{a+b}{2}] \cup [\frac{a+b}{2}, b]\} \backslash \{[a, b]\}$
10: **end for**
11: **Output**: location $\widehat{t_\varepsilon} \leftarrow \arg\max_{t_i} W(t_i)$ and its value $W\left(\widehat{t_\varepsilon}\right)$

---

More formally, let $\varepsilon$ be the required precision, the *only* given argument of the algorithm. For any $0 \leq a < b \leq 1$, the interval $[a, b]$ is associated with an upper bound $B_{[a,b]}$ defined by

$$B_{[a,b]} \triangleq \max(W(a), W(b)) + \eta_\varepsilon(b-a), \quad \text{where} \quad \forall \delta > 0 \text{ s.t. } \varepsilon\delta \leq \frac{1}{2}, \quad \eta_\varepsilon(\delta) \triangleq \sqrt{\frac{5\delta}{2} \ln\left(\frac{2}{\varepsilon\delta}\right)}.$$

OOB keeps track of a set $\mathcal{I}$ of intervals $[a, b]$ with $W$ already being sampled at $a$ and $b$. The algorithm first samples $W(1), W(1) \sim \mathcal{N}(0, 1)$, in order to initialize the set $\mathcal{I}$ to the singleton $\{[0, 1]\}$. Then, OOB keeps splitting the interval $I \in \mathcal{I}$ associated with the highest upper bound $B_I$ quam necessarium.

## 3 Guarantees: OOB is correct and sample-efficient

Let $M \triangleq \sup_{t \in [0,1]} W(t)$ be the maximum of the Brownian motion, $\hat{t}_\varepsilon$ the output of OOB called with parameter $\varepsilon > 0$, and $N_\varepsilon$ the number of Brownian evaluations performed until OOB terminates. All are random variables that depend on the Brownian motion $W$. We now voice our main result.

**Theorem 1.** *There exists a constant $c > 0$ such that for all $\varepsilon < 1/2$,*

$$\mathbb{P}\big[M - W(\hat{t}_\varepsilon) > \varepsilon\big] \leq \varepsilon \quad \text{and} \quad \mathbb{E}[N_\varepsilon] \leq c\log^2(1/\varepsilon).$$

The first inequality quantifies the *correctness* of our estimator $\widehat{M}_\varepsilon = W(\hat{t}_\varepsilon)$. Given a realization of the Brownian motion, our OOB is deterministic. The only source of randomness comes from the realization of the Brownian. Therefore, being correct means that among all possible realizations of the Brownian motion, there is a subset of measure at least $1 - \varepsilon$ on which OOB outputs an estimate $\widehat{M}_\varepsilon$ which is at most $\varepsilon$-away from the true maximum. Such guarantee is called *probably approximately correct* (PAC). The second inequality quantifies *performance*. We claim that the expectation (over $W$) of the number of samples that OOB needs to optimize this realization with precision $\varepsilon$ is $\mathcal{O}\big(\log^2(1/\varepsilon)\big)$.

**Corollary 1.** *We get the classic $(\delta, \varepsilon)$-PAC guarantee easily. For any $\delta > 0$ and $\varepsilon > 0$, choose $\varepsilon' = \min(\delta, \varepsilon)$ and apply Theorem 1 for $\varepsilon'$ from which we get $\mathbb{P}\big[M - W(\hat{t}_\varepsilon) > \varepsilon'\big] \leq \varepsilon'$ which is stronger than $\mathbb{P}\big[M - W(\hat{t}_\varepsilon) > \varepsilon\big] \leq \delta$. Similarly, $\mathbb{E}[N_{\varepsilon'}] \leq c\log^2(1/\varepsilon') \leq 4c(\log(1/\varepsilon) + \log(1/\delta))^2$.*

**Remark 1.** *Our PAC guarantee is actually stronger than stated in Theorem 1. Indeed, the PAC guarantee analysis can be done conditioned on the collected function evaluations and get*

$$\mathbb{P}\big[M - W(\hat{t}_\varepsilon) > \varepsilon | W(t_1), ..., W(t_{N_\varepsilon})\big] \leq \varepsilon,$$

*from which taking the expectation on both sides gives the first part of Theorem 1. This means that the unfavorable cases, i.e., the Brownian realizations for which $\big|M - \widehat{M}_\varepsilon\big| > \varepsilon$, are not concentrated on some subsets of Brownian realizations matching some evaluations in $t_1, ..., t_{N_\varepsilon}$. In other words, the PAC guarantee also holds when restricted to the Brownian realizations matching the evaluations in $t_1, ..., t_{N_\varepsilon}$ only. This is possible because $N_\varepsilon$ is not fixed but depends on the evaluations done by OOB.*

One difference from the result of Calvin et al. (2017) is that theirs is with respect to the $\mathcal{L}_p$ norm. For their algorithm, they prove that with $n$ samples it returns $\hat{t}_n \in [0, 1]$ such that

$$\forall r > 1, p > 1, \quad \exists c_{r,p}, \quad \mathbb{E}\Big[\big|M - W(\hat{t}_n)\big|^p\Big]^{1/p} \leq c_{r,p}/n^r.$$

To express their result in the same formalism as ours, we first choose to achieve accuracy $\varepsilon^2$ and compute the number of samples $n_{\varepsilon^2}$ needed to achieve it. Then, for $p = 1$, we apply Markov inequality and get that for all $r > 1$ there exists $c_{r,1}$ such that

$$\mathbb{P}\big[M - W(\hat{t}_{n_{\varepsilon^2}}) > \varepsilon\big] \leq \varepsilon \quad \text{and} \quad N_\varepsilon \leq c_{r,1}/\varepsilon^{1/r}.$$

On the other hand, in our Theorem 1 we give a *poly-logarithmic* bound for the sample complexity and we are in the business because this is better than any polynomial rate.

## 4 Analysis and the proof of the main theorem

We provide a proof of the main result. Let $\mathcal{I}_{\text{fin}}$ be the set $\mathcal{I}$ of intervals tracked by OOB when it finishes. We define an event $\mathcal{C}$ such that for any interval $I$ of the form $I = [k/2^h, (k+1)/2^h]$ with $k$ and $h$ being two integers where $0 \leq k < 2^h$, the process $W$ is lower than $B_I$ on the interval $I$.

**Definition 1.** *Event $\mathcal{C}$ is defined as*

$$\mathcal{C} \triangleq \bigcap_{h=0}^{\infty} \bigcap_{k=0}^{2^h-1} \left\{ \sup_{t\in[k/2^h,(k+1)/2^h]} W(t) \leq B_{[k/2^h,(k+1)/2^h]} \right\}.$$

Event $\mathcal{C}$ is a proxy for the Lipschitz condition on $W$ for the pseudo-distance $d(x,y) = \sqrt{|y-x|\ln(1/|y-x|)}$ because $\eta_\varepsilon(\delta)$ scales with $\delta$ as $\sqrt{\delta \ln(1/\delta)}$. We show that it holds with high probability. To show it, we make use of the *Brownian bridge* which is the process $\mathrm{Br}(t) \triangleq (W(t)|W(1)=0)$. Lemma 1 follows from the Markov property of the Brownian combined with a bound on the law of the maximum of $\mathrm{Br}(t)$ to bound the probability $\mathbb{P}[\sup_{t\in I} W(t) \geq B_I]$ for any $I$ of the form $[k/2^h,(k+1)/2^h]$ and a union bound over all these intervals.

**Lemma 1.** *For any $\varepsilon$, event $\mathcal{C}$ from Definition 1 holds with high probability. In particular,*

$$\mathbb{P}[\mathcal{C}^c] \leq \varepsilon^5.$$

*Proof.* For any interval $I$,

$$\begin{aligned}
B_I &= \max(W(a),W(b)) + \eta_\varepsilon(b-a) && \text{(by definition of } B_I) \\
&= \frac{W(a)+W(b)}{2} + \frac{|W(a)-W(b)|}{2} + \eta_\varepsilon(b-a) && (\max(x,y)=(x+y+|x-y|)/2) \\
&\geq \frac{W(a)+W(b)}{2} + \sqrt{\left(\frac{W(a)-W(b)}{2}\right)^2 + (\eta_\varepsilon(b-a))^2} && (\forall x,y>0, (x+y)^2 \geq x^2+y^2).
\end{aligned}$$

We now define

$$t_{h,k} \triangleq \frac{k}{2^h}, \qquad \Delta_{h,k} \triangleq \frac{W(t_{h,k})-W(t_{h,k+1})}{2}, \qquad \eta_h \triangleq \eta_\varepsilon(b-a),$$

and

$$\mathcal{A}_{h,k} \triangleq \left\{ \sup_{t\in[k/2^h,(k+1)/2^h]} W(t) > B_{[k/2^h,(k+1)/2^h]} \right\}.$$

First, for any $a<b$, the law of the maximum of a Brownian bridge gives us

$$\forall x \geq \max(W(a),W(b)) : \mathbb{P}\left[ \sup_{t\in[a,b]} W(t) > x \,\middle|\, W(a)=W_a, W(b)=W_b \right] = \exp\left(-\frac{2(x-W_a)(x-W_b)}{b-a}\right).$$

Combining it with the definition of $\mathcal{A}_{h,k}$ and the first inequality of the proof we get

$$\begin{aligned}
&\mathbb{P}\big[\mathcal{A}_{h,k}\big|W(t_h,k),W(t_h,k+1)\big] \\
&\leq \exp\left(-\frac{2\left(\frac{W(t_{h,k+1})-W(t_{h,k})}{2}+\sqrt{\Delta_{h,k}^2+\eta_h^2}\right)\left(\frac{W(t_{h,k})-W(t_{h,k+1})}{2}+\sqrt{\Delta_{h,k}^2+\eta_h^2}\right)}{t_{h,k+1}-t_{h,k}}\right) \\
&= \exp\left(-2^{h+1}\left(\sqrt{\Delta_{h,k}^2+\eta_h^2}-\Delta_{h,k}\right)\left(\sqrt{\Delta_{h,k}^2+\eta_h^2}+\Delta_{h,k}\right)\right) \\
&= \exp\left(-2^{h+1}\eta_h^2\right) = \exp\left(-2^{h+1}\frac{5}{2\cdot 2^h}\ln\left(\frac{2^h}{\varepsilon}\right)\right) = \left(\frac{\varepsilon}{2^h}\right)^5.
\end{aligned}$$

By definition, $\mathcal{C} \triangleq \bigcap_{h=0}^{\infty}\bigcap_{k=0}^{2^h-1}\mathcal{A}_{h,k}^c = \bigcup_{h=0}^{\infty}\bigcup_{k=0}^{2^h-1}\mathcal{A}_{h,k}$. By union bound on all $\mathcal{A}_{h,k}$ we get

$$\mathbb{P}[\mathcal{C}^c] \leq \sum_{h\geq 1}\sum_{k=0}^{2^h-1}\mathbb{P}[\mathcal{A}_{h,k}] \leq \sum_{h\geq 1}\sum_{k=0}^{2^h-1}\left(\frac{\varepsilon}{2^h}\right)^5 \leq \sum_{h\geq 1}\frac{\varepsilon^5}{2^{4h}} \leq \varepsilon^5. \qquad \square$$

Lemma 1 is useful for two reasons. As we bound the sample complexity on event $\mathcal{C}$ and the complementary event in two different ways, we can use Lemma 1 to combine the two bounds to prove Proposition 2 in the end. We also use a weak version of it, bounding $\varepsilon^5$ by $\varepsilon$ to prove our PAC guarantee. For this purpose, we combine the definition of $\mathcal{C}$ with the termination condition of `OOB` to get that under event $\mathcal{C}$, the best point $\widehat{M}_\varepsilon$ so far, is close to the maximum $M$ of the Brownian up to $\varepsilon$. Since $\mathcal{C}$ holds with high probability, we have the following PAC guarantee which is the first part of the main theorem.

**Proposition 1.** *The estimator $\widehat{M}_\varepsilon = W\left(\widehat{t}_\varepsilon\right)$ is probably approximately correct with*

$$\mathbb{P}\left[M - \widehat{M}_\varepsilon > \varepsilon\right] \leq \varepsilon.$$

*Proof.* Let $I_{\text{next}} = [a,b]$ be the interval that the algorithm would split next, if it was not terminated. Since the algorithm only splits the interval with the highest upper bound then $B_{\text{next}} = \sup_{I \in \mathcal{I}_{\text{fin}}} B_I$. Also let $I_{\max} \in \mathcal{I}_{\text{fin}}$ be one of the intervals where a maximum is reached, $t_{\max} \in \arg\max_{t \in [0,1]} W(t) \triangleq M$ and $t_{\max} \in I_{\max}$. Then, on event $\mathcal{C}$,

$$M \leq B_{I_{\max}} \leq B_{I_{\text{next}}} = \max(W(a), W(b)) + \eta_\varepsilon(b - a).$$

Since the algorithm terminated, we have that $\eta_\varepsilon(b - a) \leq \varepsilon$ and therefore,

$$\max(W(a), W(b)) \geq M - \varepsilon,$$

which combined with Lemma 1 finishes the proof as $\varepsilon^5 \leq \varepsilon$. $\qquad\square$

In fact, Proposition 1 is the easy-to-obtain part of the main theorem. We are now left to prove that the number of samples needed to achieve this PAC guarantee is low. As the next step, we define the near-optimality property. A point $t$ is said to be $\eta$-near-optimal when its value $W(t)$ is close to the maximum $M$ of the Brownian motion up to $\eta$. Check out the precise definition below.

**Definition 2.** *When an $(h, k, \eta)$ verifies $W\left(\frac{k}{2^h}\right) \geq M - \eta$, we say that the point $t = (k/2^h)$ is $\eta$-near-optimal. We define $\mathcal{N}_h(\eta)$ as the number of $\eta$-near-optimal points among $\{0/2^h, 1/2^h, ..., 2^h/2^h\}$,*

$$\mathcal{N}_h(\eta) \triangleq \left|\left\{k \in 0, ..., 2^h, \text{ such that } W\left(\frac{k}{2^h}\right) \geq M - \eta\right\}\right|.$$

Notice that $\mathcal{N}_h(\eta)$ is a random variable that depends on a particular realization of the Brownian motion. The reason why we are interested in the number of near-optimal points is that the points the algorithm will sample are $\eta_\varepsilon\left(1/2^h\right)$-near-optimal. Since we use the principle of optimism in face of uncertainty, we consider the upper bound of the Brownian motion and sample where this upper bound is the largest. If our bounds on $W$ hold, i.e., under event $\mathcal{C}$, then any interval $I$ with optimistic bound $B_I < M$ is never split by the algorithm. This is true when $\mathcal{C}$ holds because if the maximum of $W$ is reached in $I_{\max}$, then $B_{I_{\max}} \geq M > B_I$ which shows that $I_{\max}$ is always chosen over $I$. Therefore, a necessary condition for an interval $[a, b]$ to be split is that $\max(W(a), W(b)) \geq M - \eta$ which means that either $a$ or $b$ or both are $\eta$-near-optimal which is the key point of Lemma 2.

**Lemma 2.** *Under event $\mathcal{C}$, the number of evaluated points $N_\varepsilon$ by the algorithm verifies*

$$N_\varepsilon \leq 2 \sum_{h=0}^{h_{\max}} \mathcal{N}_h\left(\eta_\varepsilon\left(1/2^h\right)\right), \text{with } h_{\max} \text{ being the smallest } h \text{ such that } \eta_\varepsilon\left(1/2^h\right) \leq \varepsilon.$$

Lemma 2 explicitly links the near-optimality from Definition 2 with the number of samples $N_\varepsilon$ performed by OOB before it terminates. Here, we use the optimism-in-face-of-uncertainty principle which can be applied to any function. In particular, we define a high-probability event $\mathcal{C}$ under which the number of samples is bounded by the number of near-optimal points $\mathcal{N}_h(\eta_h)$ for all $h \leq h_{\max}$.

*Proof.* Let $I = [a, b]$ be an interval of $\mathcal{I}_t$ such that $\max(W(a), W(b)) + \eta_\varepsilon(b - a) < M$. Let $I_{\text{next}} \in \mathcal{I}_t$ be the interval that the algorithm would split after $t$ function evaluations.. Since the algorithm only splits the interval with the highest upper bound, then $B_{I_{\text{next}}} = \sup_{I \in \mathcal{I}_t} B_I$. Moreover, if we let $I_{\max} \in \mathcal{I}_t$ be one of the intervals where a maximum is reached, $t_{\max} \in \arg\max_{t \in [0,1]} W(t) \triangleq M$ and $t_{\max} \in I_{\max}$, then on event $\mathcal{C}$,

$$\max(W(a), W(b)) + \eta_\varepsilon(b - a) \triangleq B_I < M \leq B_{I_{\max}} \leq B_{I_{\text{next}}}.$$

Therefore, under $\mathcal{C}$, a necessary condition for an interval $I = [a, b]$ to be split during the execution of OOB is that $\max(W(a), W(b)) \geq M - \eta_\varepsilon(b - a)$, which means that either $a$ or $b$ or both are $\eta_\varepsilon(b - a)$-near-optimal. From the termination condition of the algorithm, we know that any interval that is satisfying $I = [k/2^h, (k + 1)/2^h]$ with $h \geq h_{\max}$ will not be split during the execution.[1] Therefore, another necessary condition for an interval $I = [a, b]$ to be split during the execution is that $b - a > 1/2^{h_{\max}}$. Writing $\eta_h \triangleq \eta_\varepsilon\left(1/2^h\right)$, we deduce from these two necessary conditions that

$$N \leq \sum_{h=0}^{h_{\max}} \sum_{k=0}^{2^h-1} \mathbf{1}\left\{\frac{k}{2^h} \text{ or } \frac{k+1}{2^h} \text{ is } \eta_h\text{-near-optimal}\right\}$$

$$\leq \sum_{h=0}^{h_{\max}} \sum_{k=0}^{2^h-1} \mathbf{1}\left\{\frac{k}{2^h} \text{ is } \eta_h\text{-near-optimal}\right\} + \mathbf{1}\left\{\frac{k+1}{2^h} \text{ is } \eta_h\text{-near-optimal}\right\}$$

$$\leq 2 \sum_{h=0}^{h_{\max}} \sum_{k=0}^{2^h} \mathbf{1}\left\{\frac{k}{2^h} \text{ is } \eta_h\text{-near-optimal}\right\} = 2 \sum_{h=0}^{h_{\max}} \mathcal{N}_h(\eta_h). \qquad \square$$

We now prove a property *specific to $W$* by bounding the number of near-optimal points of the *Brownian motion* in expectation. We do it by rewriting it as two Brownian *meanders* (Durrett et al., 1977), both starting at the maximum of the Brownian, one going backward and the other one forward with the Brownian meander $W^+$ defined as

$$\forall t \in [0,1] \quad W^+(t) \triangleq \frac{|W(\tau + t(1-\tau))|}{\sqrt{1-\tau}}, \quad \text{where } \tau \triangleq \sup\{t \in [0,1] : W(t) = 0\}.$$

We use that the Brownian meander $W^+$ can be seen as a Brownian motion conditioned to be positive (Денисов, 1983). This is the main ingredient of Lemma 3.

**Lemma 3.** *For any $h$ and $\eta$, the expected number of near-optimal points is bounded as*

$$\mathbb{E}[\mathcal{N}_h(\eta)] \leq 6\eta^2 2^h.$$

This lemma answers a question raised by Munos (2011): *What is the near-optimality dimension of the Brownian motion?* We set $\eta_h \triangleq \eta_\varepsilon(1/2^h)$. In dimension one with the pseudo-distance $\ell(x,y) = \eta_\varepsilon(|y-x|)$, the near-optimality dimension measures the rate of increase of $\mathcal{N}_h(\eta_h)$, the number of $\eta_h$-near-optimal points in $[0,1]$ of the form $k/2^h$. In Lemma 3, we prove that in expectation, this number increases as $\mathcal{O}(\eta_h^2 2^h) = \mathcal{O}(\log(1/\varepsilon))$, which is constant with respect to $h$. This means that for a given $\varepsilon$, there is a metric under which the Brownian is Lipschitz with probability at least $1 - \varepsilon$ and has a near-optimality dimension $d = 0$ with $C = \mathcal{O}(\log(1/\varepsilon))$.

The final sample complexity bound is essentially constituted by one $\mathcal{O}(\log(1/\varepsilon))$ term coming from the standard DOO error for deterministic function optimization and another $\mathcal{O}(\log(1/\varepsilon))$ term because we need to adapt our pseudo-distance $\ell$ to $\varepsilon$ such that the Brownian is $\ell$-Lipschitz with probability $1 - \varepsilon$. The product of the two gives the final sample complexity bound of $\mathcal{O}(\log^2(1/\varepsilon))$.

*Proof of Lemma 3.* We denote by $W$, the Brownian motion whose maximum $M$ is first hit at the point defined as $t_M = \inf\{t \in [0,1]; W(t) = M\}$ and $B^+$ a Brownian meander (Durrett et al., 1977). We also define

$$B_0^+(t) \triangleq \frac{M - W(t_M - t \cdot t_M)}{\sqrt{t_M}} \quad \text{and} \quad B_1^+(t) \triangleq \frac{M - W(t_M + t(1-t_M))}{\sqrt{1-t_M}}.$$

If $\stackrel{\mathcal{L}}{=}$ denotes the equality in distribution, then Theorem 1 of Денисов (1983) asserts that

$$B^+ \stackrel{\mathcal{L}}{=} B_0^+ \stackrel{\mathcal{L}}{=} B_1^+ \text{ and } t_M \text{ is independent from both } B_0^+ \text{ and } B_1^+.$$

We upper-bound the expected number of $\eta$-near-optimal points for any integer $h \geq 0$ and any $\eta > 0$,

$$\mathbb{E}[\mathcal{N}_h(\eta)] = \mathbb{E}\left[\sum_{k=0}^{2^h} \mathbf{1}\left\{W\left(\frac{k}{2^h}\right) > M - \eta\right\}\right] = \sum_{k=0}^{2^h} \mathbb{E}\left[\mathbf{1}\left\{W\left(\frac{k}{2^h}\right) > M - \eta\right\}\right]$$

$$= \sum_{k=0}^{2^h} \mathbb{E}\left[\mathbf{1}\left\{\left\{W\left(\frac{k}{2^h}\right) > M - \eta \cap \frac{k}{2^h} \leq t_M\right\} \cup \left\{W\left(\frac{k}{2^h}\right) > M - \eta \cap \frac{k}{2^h} > t_M\right\}\right\}\right]$$

$$= \sum_{k=0}^{2^h} \mathbb{E}\left[\mathbf{1}\left\{B_0^+\left(1 - \frac{k}{t_M 2^h}\right) < \frac{\eta}{\sqrt{t_M}} \cap \frac{k}{2^h} \leq t_M\right\}\right] + \sum_{k=0}^{2^h} \mathbb{E}\left[\mathbf{1}\left\{B_1^+\left(\frac{k/2^h - t_M}{1 - t_M}\right) < \frac{\eta}{\sqrt{1-t_M}} \cap \frac{k}{2^h} > t_M\right\}\right].$$

If $X$ and $Y$ are independent then for any function $f$,

$$\mathbb{E}[f(X,Y)] = \mathbb{E}[\mathbb{E}[f(X,Y)|Y]] \qquad \text{(law of total expectation)}$$
$$\leq \sup_y \mathbb{E}[f(X,y)|Y = y] \qquad \text{(for any } Z, \mathbb{E}[Z] \leq \sup_w Z(w)\text{)}$$
$$= \sup_y \mathbb{E}[f(X,y)]. \qquad \text{(because } X \text{ and } Y \text{ are independent)}$$

Since $t_M$ is independent from $B_0^+$ and $B_1^+$, then using the above with $X = (B_0^+, B_1^+), Y = t_M$, and

$$f : (x_0, x_1), y \to \sum_{k=0}^{2^h} \left( \mathbf{1}\left\{ x_0\left(1 - \frac{k}{t_M 2^h}\right) < \frac{\eta}{\sqrt{y}} \cap \frac{k}{2^h} \leq y \right\} + \mathbf{1}\left\{ x_1\left(\frac{k/2^h - t_M}{1 - y}\right) < \frac{\eta}{\sqrt{1-y}} \cap \frac{k}{2^h} > y \right\} \right),$$

we can claim that

$$\mathbb{E}[\mathcal{N}_h(\eta)] = \mathbb{E}[f(X, Y)] \leq \sup_{u \in [0,1]} \mathbb{E}[f(X, u)]$$

$$\leq \sup_{u \in [0,1]} \left\{ \sum_{k=0}^{2^h} \mathbb{E}\left[ \mathbf{1}\left\{ B_0^+\left(1 - \frac{k}{u2^h}\right) < \frac{\eta}{\sqrt{u}} \cap \frac{k}{2^h} \leq u \right\} \right] \right\}$$

$$+ \sup_{u \in [0,1]} \left\{ \sum_{k=0}^{2^h} \mathbb{E}\left[ \mathbf{1}\left\{ B_1^+\left(\frac{k/2^h - u}{1 - u}\right) < \frac{\eta}{\sqrt{1 - u}} \cap \frac{k}{2^h} > u \right\} \right] \right\}$$

$$= \sup_{u \in [0,1]} \left\{ \sum_{k=0}^{\lfloor u2^h \rfloor} \mathbb{P}\left[ B_0^+\left(1 - \frac{k}{u2^h}\right) < \frac{\eta}{\sqrt{u}} \right] \right\} + \sup_{u \in [0,1]} \left\{ \sum_{k=\lceil u2^h \rceil}^{2^h} \mathbb{P}\left[ B_1^+\left(\frac{k/2^h - u}{1 - u}\right) < \frac{\eta}{\sqrt{1 - u}} \right] \right\}$$

$$= 2 \sup_{u \in [0,1]} \left\{ \sum_{k=0}^{\lfloor u2^h \rfloor} \mathbb{P}\left[ B_0^+\left(1 - \frac{k}{u2^h}\right) < \frac{\eta}{\sqrt{u}} \right] \right\} = 2 \sup_{u \in [0,1]} \{\alpha_1 + \alpha_2 + \alpha_3 + \alpha_4\},$$

with $\alpha_1 = \sum_{k=0}^{\lfloor 2^h \eta^2 \rfloor} \mathbb{P}\left[ B_0^+\left(1 - \frac{k}{u2^h}\right) < \frac{\eta}{\sqrt{u}} \right]$, $\quad \alpha_2 = \sum_{k=\lceil 2^h \eta^2 \rceil}^{\lfloor \frac{u2^h}{2} \rfloor} \mathbb{P}\left[ B_0^+\left(1 - \frac{k}{u2^h}\right) < \frac{\eta}{\sqrt{u}} \right]$

$\alpha_3 = \sum_{k=\lceil \frac{u2^h}{2} \rceil}^{\lfloor u2^h \rfloor - \lceil 2^h \eta^2 \rceil} \mathbb{P}\left[ B_0^+\left(1 - \frac{k}{u2^h}\right) < \frac{\eta}{\sqrt{u}} \right]$, and $\alpha_4 = \sum_{k=\lfloor u2^h \rfloor - \lfloor 2^h \eta^2 \rfloor}^{\lfloor u2^h \rfloor} \mathbb{P}\left[ B_0^+\left(1 - \frac{k}{u2^h}\right) < \frac{\eta}{\sqrt{u}} \right]$.

Since a probability is always upper-bounded by 1, we bound $\alpha_1$ and $\alpha_4$ both by $\eta^2 2^h$, to get that $\alpha_1 + \alpha_4 \leq 2\eta^2 2^h$. We now bound the remaining probabilities appearing in the above expression by integrating over the distribution function of Brownian meander (Durrett et al., 1977, Equation 1.1),

$$\mathbb{P}[B_0^+(t) < x] = 2\sqrt{2\pi} \int_0^x \frac{y \exp\left(-y^2/(2t)\right)}{t\sqrt{2\pi t}} \int_0^y \frac{\exp\left(-w^2/(2(1-t))\right)}{\sqrt{2\pi(1-t)}} \, dw \, dy$$

$$\leq \frac{2}{t\sqrt{2\pi t(1 - t)}} \int_0^x y^2 \exp\left(-\frac{y^2}{2t}\right) dy$$

$$\leq \frac{2x^3}{3t\sqrt{2\pi t(1-t)}} \leq \frac{2x^3}{3t(1-t)\sqrt{2\pi t(1-t)}} = \frac{2}{3\sqrt{2\pi}}\left(\frac{x}{\sqrt{t(1-t)}}\right)^3,$$

where the first two inequalities are obtained by upperbounding the terms $\exp(\cdot)$ by one. Now, we use the above bound to bound $\alpha_2 + \alpha_3$,

$$\alpha_2 + \alpha_3 = \sum_{k=\lceil 2^h \eta^2 \rceil}^{\lfloor \frac{u2^h}{2} \rfloor} \mathbb{P}\left[ B_0^+\left(1 - \frac{k}{u2^h}\right) < \frac{\eta}{\sqrt{u}} \right] + \sum_{k=\lceil \frac{u2^h}{2} \rceil}^{\lfloor u2^h \rfloor - \lceil 2^h \eta^2 \rceil} \mathbb{P}\left[ B_0^+\left(1 - \frac{k}{u2^h}\right) < \frac{\eta}{\sqrt{u}} \right]$$

$$\leq \sum_{k=\lceil 2^h \eta^2 \rceil}^{\lfloor \frac{u2^h}{2} \rfloor} \frac{2}{3\sqrt{2\pi}}\left(\frac{\frac{\eta}{\sqrt{u}}}{\sqrt{1 - \frac{k}{u2^h}}\sqrt{\frac{k}{u2^h}}}\right)^3 + \sum_{k=\lceil \frac{u2^h}{2} \rceil}^{\lfloor u2^h \rfloor - \lceil 2^h \eta^2 \rceil} \frac{2}{3\sqrt{2\pi}}\left(\frac{\frac{\eta}{\sqrt{u}}}{\sqrt{1 - \frac{k}{u2^h}}\sqrt{\frac{k}{u2^h}}}\right)^3$$

$$\leq \sum_{k=\lceil 2^h \eta^2 \rceil}^{\lfloor \frac{u2^h}{2} \rfloor} \frac{1}{6\sqrt{\pi}}\left(\frac{\frac{\eta}{\sqrt{u}}}{\sqrt{\frac{k}{u2^h}}}\right)^3 + \sum_{k=\lceil \frac{u2^h}{2} \rceil}^{\lfloor u2^h \rfloor - \lceil 2^h \eta^2 \rceil} \frac{1}{6\sqrt{\pi}}\left(\frac{\frac{\eta}{\sqrt{u}}}{\sqrt{1 - \frac{k}{u2^h}}}\right)^3$$

$$\leq \sum_{k=\lceil 2^h \eta^2 \rceil}^{\lfloor \frac{u2^h}{2} \rfloor} \frac{1}{6\sqrt{\pi}}\left(\frac{\frac{\eta}{\sqrt{u}}}{\sqrt{\frac{k}{u2^h}}}\right)^3 + \sum_{k=\lceil 2^h \eta^2 \rceil}^{\lfloor u2^h \rfloor - \lceil \frac{u2^h}{2} \rceil} \frac{1}{6\sqrt{\pi}}\left(\frac{\frac{\eta}{\sqrt{u}}}{\sqrt{\frac{\lfloor u2^h \rfloor}{u2^h} + \frac{k}{u2^h}}}\right)^3.$$

Changing the indexation as $k = -k' + \lfloor u2^h \rfloor$, we get

$$\alpha_2 + \alpha_3 \leq \frac{\left(2^h \eta^2\right)^{3/2}}{6\sqrt{\pi}} \left( \sum_{k=\lceil 2^h \eta^2 \rceil}^{\lfloor \frac{u2^h}{2} \rfloor} \frac{1}{k^{3/2}} + \sum_{k=\lceil 2^h \eta^2 \rceil}^{\lfloor u2^h \rfloor - \lceil \frac{u2^h}{2} \rceil} \frac{1}{k^{3/2}} \right)$$

$$\leq \frac{\left(2^h \eta^2\right)^{3/2}}{3\sqrt{\pi}} \sum_{k=\lceil 2^h \eta^2 \rceil}^{\infty} \frac{1}{k^{3/2}} \leq \frac{\left(2^h \eta^2\right)^{3/2}}{3\sqrt{\pi}} \frac{3}{\sqrt{\lceil 2^h \eta^2 \rceil}} \leq \frac{1}{\sqrt{\pi}} \eta^2 2^h \leq \eta^2 2^h,$$

where in the last line we used that for any $k_0 \geq 1$,

$$\sum_{k=k_0}^{\infty} \frac{1}{k^{3/2}} \leq \frac{1}{k_0^{3/2}} + \sum_{k=k_0+1}^{\infty} \int_{k-1}^{k} \frac{1}{u^{3/2}} \, \mathrm{d}u = \frac{1}{k_0^{3/2}} + \int_{k_0}^{\infty} \frac{1}{u^{3/2}} \, \mathrm{d}u = \frac{1}{k_0^{3/2}} + \frac{2}{\sqrt{k_0}} \leq \frac{3}{\sqrt{k_0}}.$$

We finally have

$$\forall u, \quad \alpha_1 + \alpha_2 + \alpha_3 + \alpha_4 \leq 3\eta^2 2^h$$

and therefore

$$\mathbb{E}[\mathcal{N}_h(\eta)] \leq 6\eta^2 2^h. \qquad \square$$

To conclude the analysis, we put Lemma 2 and Lemma 3 together in order to bound the sample complexity conditioned on event $\mathcal{C}$.

**Lemma 4.** *There exists $c > 0$ such that for all $\varepsilon \leq 1/2$, $\mathbb{E}[N_\varepsilon \mid \mathcal{C}] \leq c \log^2(1/\varepsilon)$.*

*Proof.* By definition of $h_{\max}$,

$$\varepsilon^2 \leq \frac{5}{2 \times 2^{h_{\max}}} \ln\left( \frac{2^{h_{\max}}}{\varepsilon} \right) \leq \frac{5}{2} \frac{\sqrt{2^{h_{\max}}/\varepsilon}}{2^{h_{\max}}},$$

from which we deduce that

$$\sqrt{\varepsilon} \varepsilon^2 \leq \frac{5}{2 \times 2^{h_{\max}/2}} \quad \text{and therefore} \quad 2^{h_{\max}} \leq \frac{25}{4\varepsilon^5},$$

which gives us an upper bound on $h_{\max}$,

$$h_{\max} \leq \frac{\ln(25/4)}{\ln 2} + \frac{5 \ln(1/\varepsilon)}{\ln 2} = \mathcal{O}(\log(1/\varepsilon)).$$

Furthermore, using Lemma 1, we get that for any $\varepsilon \leq 1/2$,

$$\mathbb{E}[\mathcal{N}_h(\eta_h)] = \mathbb{E}[\mathcal{N}_h(\eta_h)\mid \mathcal{C}]\mathbb{P}[\mathcal{C}] + \mathbb{E}[\mathcal{N}_h(\eta_h)\mid \mathcal{C}^c]\mathbb{P}[\mathcal{C}^c] \geq \mathbb{E}[\mathcal{N}_h(\eta_h)\mid \mathcal{C}](1 - \varepsilon) \geq \frac{\mathbb{E}[\mathcal{N}_h(\eta_h)\mid \mathcal{C}]}{2}.$$

We now use Lemma 2 and Lemma 3 to get

$$\mathbb{E}[N_\varepsilon \mid \mathcal{C}] \leq 2 \sum_{h=0}^{h_{\max}} \mathbb{E}\left[ \mathcal{N}_h\left( \sqrt{\frac{5}{2} \frac{\ln(2^{h+1}/\varepsilon)}{2^h}} \right) \Big| \mathcal{C} \right] \leq 60 \sum_{h=0}^{h_{\max}} \ln\left( 2^{h+1}/\varepsilon \right)$$

$$= 60 \sum_{h=1}^{h_{\max}+1} \left( \ln(1/\varepsilon) + h \ln 2 \right) = \mathcal{O}\left( h_{\max}^2 + h_{\max} \log(1/\varepsilon) \right) = \mathcal{O}\left( \log^2(1/\varepsilon) \right). \qquad \square$$

We also bound the sample complexity on $\mathcal{C}^c$, the event complementary to $\mathcal{C}$, by the total number of possible intervals $[k/2^h, (k+1)/2^h]$ with $\eta_\varepsilon(1/2^h) \leq \varepsilon$. Then, we combine it with Lemma 1, Lemma 2, and Lemma 3 to get Proposition 2 which is the second part of the main theorem.

**Proposition 2.** *There exists $c > 0$ such that for all $\varepsilon < 1/2$,*

$$\mathbb{E}[N_\varepsilon] \leq c \log^2(1/\varepsilon).$$

*Proof.* From the law of total expectation,

$$\mathbb{E}[N_\varepsilon] = \mathbb{E}[N_\varepsilon \mid \mathcal{C}]\mathbb{P}[\mathcal{C}] + \mathbb{E}[N_\varepsilon \mid \mathcal{C}^c]\mathbb{P}[\mathcal{C}^c] \leq \mathbb{E}[N_\varepsilon \mid \mathcal{C}] + \varepsilon^5 \mathbb{E}[N_\varepsilon \mid \mathcal{C}^c].$$

By Lemma 4, we have that $\mathbb{E}[N_\varepsilon \,|\, \mathcal{C}\,] = \mathcal{O}\big(\log^2(1/\varepsilon)\big)$. Now let $h_{\max}$ be the maximum $h$ at which points are evaluated. Then,

$$N_\varepsilon \leq \sum_{h=0}^{h_{\max}} 2^h = 2^{h_{\max}+1} - 1 \leq 2^{h_{\max}+1}.$$

As in the proof of Lemma 4, we get $2^{h_{\max}+1} = \mathcal{O}\big(\varepsilon^{-5}\big)$. We finally obtain that

$$\mathbb{E}[N_\varepsilon] \leq \mathcal{O}\big(\log^2(1/\varepsilon)\big) + \mathcal{O}(1) = \mathcal{O}\big(\log^2(1/\varepsilon)\big). \qquad \square$$

Proposition 2 together with Proposition 1 establish the proof of Theorem 1, the holy grail.

## 5 Numerical evaluation of OOB

For an illustration, we ran a simple experiment and for different values of $\varepsilon$, we computed the average empirical sample complexity $N_\varepsilon$ on 250 independent runs that you can see on the left plot. We also plot one point for each run of OOB instead of averaging the sample complexity, to be seen on the right. The experiment indicates a linear dependence between the sample complexity and $\log^2(1/\varepsilon)$.

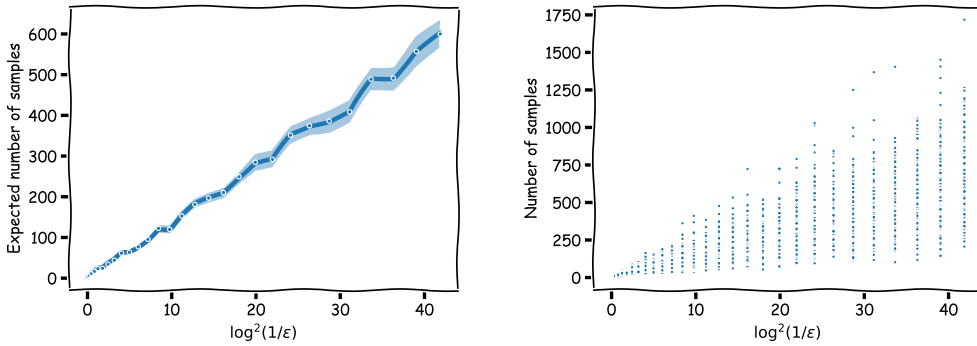

## 6 Conclusion and ideas for extensions and generalizations

We presented OOB, an algorithm inspired by DOO (Munos, 2011) that efficiently optimizes a Brownian motion. We proved that the sample complexity of OOB is of order $\mathcal{O}\big(\log^2(1/\varepsilon)\big)$ which improves over the previously best known bound (Calvin et al., 2017). As we are not aware of a lower bound for the Brownian motion optimization, we do not know whether $\mathcal{O}\big(\log^2(1/\varepsilon)\big)$ is the minimax rate of the sample complexity or if there exists an algorithm that can do even better.

What would be needed to do if we wanted to use our approach for Gaussian processes with a different kernel? The optimistic approach we took is quite general and only Lemma 3 would need additional work as this is the ingredient most specific to Brownian motion. Notice, that Lemma 3 bounds the number of near-optimal nodes of a Brownian motion in expectation. To bound the expected number of near-optimal nodes, we use the result of Денисов (1983) which is based on 2 components:

1 A Brownian motion can be rewritten as an affine transformation of a Brownian motion conditioned to be positive, translated by an (independent) time at which the Brownian motion attains its maximum.

2 A Brownian motion conditioned to be positive is a Brownian meander. It requires some additional work to prove that a Brownian motion conditioned to be positive is actually properly defined.

A similar result for another Gaussian process or its generalization of our result to a larger class of Gaussian processes would need to adapt or generalize these two items in Lemma 3. On the other hand, the adaptation or generalization of the other parts of the proof would be straightforward. Moreover, for the second item, the full law of the process conditioned to be positive is actually not needed, only the local time of the Gaussian process conditioned to be positive at points near zero.

**Acknowledgements** This research was supported by European CHIST-ERA project DELTA, French Ministry of Higher Education and Research, Nord-Pas-de-Calais Regional Council, Inria and Otto-von-Guericke-Universität Magdeburg associated-team north-European project Allocate, French National Research Agency projects ExTra-Learn (n.ANR-14-CE24-0010-01) and BoB (n.ANR-16-CE23-0003), FMJH Program PGMO with the support to this program from Criteo, a doctoral grant of École Normale Supérieure, and Maryse & Michel Grill.

## Footnotes

[1]This holds despite $\eta_\varepsilon(\cdot)$ is not always decreasing.

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
