[Reviews · NeurIPS 2018]

Reviewer 1



The authors propose a new algorithm and its analysis for finding the maximum of a realized Brownian motion. The sample complexity for epsilon optimality is shown to be log^2(1/epsilon) which is better than prior art that has a polynomial dependence on 1/\epsilon. The experiment confirms the claim sample complexity order. Overall, a very well-written paper with significant development on the problem. I have a few questions and minor suggestions. At least, I think the properties assumed on W can be listed in the supplementary material. I think the paper could improve readability and clarity even better by properly define Brownian motion (at least briefly) and then citing a good reference to learn about the Brownian motion. Q: I am not too familiar with Brownian motion. Would the bounded domain ([0,1]) for Brownian violate the assumption the increments must be independent? If we start at 0 and take the first increment at t=1, with some probability we arrive near 1. Then, the increment at t=2 would necessarily be limited since we cannot go beyond 1, and so I feel like the increment at t=2 has to be correlated with the one at t=1. I am sure there must be some resolution. What do the authors think? (*) minor comments 1. Algorithm 1: the loop must have i <- i + 1 at the end 2. Algorithm 1's Output: "and its value W(hat t_eps) <- ..." => could just leave it with ""and its value W(hat t_eps)" without the RHS. I am not sure what value "<- ..." brings in there. 3. L117 (line 117): The full proofs the ... => The full proofs "of" the 4. L220 (line 220): work as this the ingredient ... => work as this "is" the ingredient...

Reviewer 2



This paper considers a classic problem in Bayesian optimization, where we seek to maximize a one-dimensional function evaluated without noise with a Brownian motion prior. Using a novel algorithm and analysis that builds on Munos 2011 but goes substantially beyond it, it shows that an epsilon-optimal point can be found using log^2(1/epsilon) samples. This provides over the previous best-known rate for this problem. The results are quite nice from the point of view of the theoretical foundations of Bayesian optimization. Indeed, optimizing with a Brownian motion prior has been studied since the 1970s by Calvin and Zilinskas especially, and this result improves on all of those results. Thus, I feel the work is of high quality. The work is also original --- although the paper builds on work by Munos and Calvin, there is a lot that is new. The paper is also quite clear. The only problem is with the paper's significance for the NIPS community, which often wants to see meaningful empirical results and practical applicability. Unfortunately one-dimensional problems are much less common in practice than multi-dimensional ones, and Wiener process priors often don't fit the functions we optimize very well. I feel that this work *is* significant from a theory perspective, given the dearth of results on convergence rates for Bayesian optimization, and is a stepping stone to understanding rates for more practical settings. Beyond this, I only have some minor advice on clarity: Lipschitz discussion: When I read the initial parts of the paper, I found the discussion of Lipschitz continuity confusing, since the path of a Brownian motion is not differentiable, and I believe is not Lipschitz continuous in the classical sense. Here we are looking at finite meshes of decreasing size, and so after reading into the body of the paper I understood what was meant, but perhaps there is a way to write the initial discussion in a way that does not confuse readers familiar with the infinite roughness of Brownian motion's paths. Throughout, I find the word "Brownian" without "motion" or "bridge" strange. I recommend talking about a "Brownian motion" rather than a "Brownian". If you like, you could abbreviate this at BM. Typos: line 117 typo: "The full proofs the auxiliary lemmas" line 139 typo: "achieving such PAC guarantee is low." line 150 and 151: it would be more clear to write the necessary condition for splitting as max(W(a),W(b)) >= M - eta*(b-a).

Reviewer 3



The authors analyze optimizing a sample function from a Brownian motion in [0,1] with a as few "active" samples as possible. They show that they can achieve an epsilon-approximation by querying the function at log(1/epsilon) points. The latter is an improvement over prior work by Calvin et al, which shows lower than polynomial sample complexity but not logarithmic. Their main result uses a notion of near-optimality dimension proposed by Munos'11 which captures how flat is the function around its optimum: how many disjoint balls can be fit in the region of x's that are epsilon minimizers. Their main finding is to bound the near optimality dimension of a sample of a Brownian motion with high probability. Given this finding their algorithm is pretty simple: simply keep an upper confidence value for each partition of the [0,1] interval created by the points queried so far. Then query the center of the partition with higher UCB value. I think the result is interesting on the near-optimality dimension of a Brownian motion of independent interest.